# Nitrogen Use Efficiency and Partitioning of Dairy Heifers Grazing Perennial Ryegrass (*Lolium perenne* L.) or Pasture Brome (*Bromus valdivianus* Phil.) Swards during Spring



Ignacio E. Beltran [1], Daniel Tellez [2], Jaime Cabanilla [2], Oscar Balocchi [3], Rodrigo Arias [3] and Juan Pablo Keim [3,*]

1 Instituto de Investigaciones Agropecuarias, INIA Remehue, Casilla 24-O, Osorno 5290000, Chile
2 Graduate School, Faculty of Agricultural and Food Sciences, Austral University of Chile, Independencia 641, Valdivia 5110566, Chile
3 Institute of Animal Production, Faculty of Agricultural and Food Sciences, Austral University of Chile, Independencia 641, Valdivia 5110566, Chile
* Correspondence: juan.keim@uach.cl

**Abstract:** The aim of the study was to evaluate the effect of grazing *Lolium perenne* (Lp) and *Bromus valdivianus* (Bv) on the average daily weight gain (ADG) and nitrogen use efficiency (NUE) of Holstein Friesian heifers. Thirty heifers strip-grazed two pasture treatments (Lp and Bv) under a randomized complete block design (n = 3). Nutrient concentration and pasture intake were determined. Urine samples were taken, and the total volume of urine and microbial growth were estimated. Retained nitrogen (N), N intake, N excreted in feces and urine and the nitrogen use efficiency (NUE) were calculated. *Lolium perenne* showed greater WSC and ME but lower NDF than Bv, whereas crude and soluble protein were unaffected. There were no effects of species on ADG or feed conversion, and DMI was not affected by grass species, or the synthesis of microbial protein and purine derivatives. Ammonia in the rumen, urinary N and total N excreted were greater for heifers grazing Bv. In conclusion, the consumption of forage species did not alter the ADG or NUE of grazing heifers, but N partitioning was modified for heifers grazing Bv, due to the lower WSC/CP ratio compared with Lp.

**Keywords:** bromus; nitrogen excretion; environmental pollution; growing dairy cattle

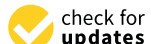



## 1. Introduction

The increasing global population has augmented the demand for animal products, such as beef and dairy products, the production of which is expected to double by 2050 compared with 2000, especially in developing countries [1]. Despite the importance of dairy cattle for global food security, the production of dairy products results in direct and indirect greenhouse gas emissions (GHGs) such as methane ($CH_4$) and nitrous oxide ($N_2O$), contributing to climate change [2,3]. One of the most important consequences of climate change will be the increase in the mean global temperature between 2.6 and 4.8 °C by the end of the 21st century (2081–2100) relative to climate conditions between 1986 and 2005 [4], leading to changes in water availability, the quality of forage and production and reproduction parameters of livestock [1,5].

Nitrogen deposited into the pasture (urine plus dung) from pasture-based dairy farming is an important source of GHG (nitrous oxide) and non-$CO_2$ (e.g., ammonia; $NH_3$) gases [6]. The environmental importance of $N_2O$ is related to its high global warming potential (265 times greater than $CO_2$) [7], while $NH_3$ is considered an air pollutant when produced through volatilization, and an indirect source of $N_2O$ emissions [8,9], contributing to global warming. Nitrogen excreted by urine and dung can represent up to 80–90% of N intake [10], reflecting that only 10–20% of ingested N is retained by grazing animals. For Chilean dairy cattle, nitrogen use efficiency (NUE) ranges between 22 and 27%, suggesting

that over 73% of ingested N can be excreted into environment [11]. The low NUE for dairy cattle has been correlated with the high N content (which often exceeds the N requirements of dairy cattle) [12] and the low fermentable carbohydrates supply from pastures [13]. Therefore, less N is captured by rumen microbes, and consequently, a high proportion of ingested N is absorbed from the rumen as $NH_3$, being converted into urea in the liver and then excreted through the urine [14]. The challenge for the dairy cattle sector is satisfying the growing demand while reducing N excretion and adapting to high temperatures and low precipitations imposed by climate change [15].

Studies evaluating these topics have been focused on improving milk performance (production and composition) and reducing N excretion from dairy cows, which is expected due to their importance in the economic stability of dairy operations [16]. Replacement heifers are considered as a key factor determining the future of dairy farms. However, they are not always prioritized [17]. The development of strategies that can maintain/improve average daily gain (ADG) and reduce nitrogen excretion from growing heifers is required for the forthcoming sustainability challenges for dairy farms.

*Bromus valdivianus* is a native species from the humid temperate region of Chile characterized by a high tolerance to restricted conditions of soil water (dry periods), mainly related to its deep root system, allowing for water from lower soil strata to be obtained [18,19]. Additionally, Bv is characterized by a high pasture availability, being similar to Lp, the most common grass species used for pasture-based dairy systems, and therefore has the potential to increase current levels of herbage production and pasture persistence in drylands [20]. Moreover, Bv shows a high content of protein (~19% CP) and fiber (~48% NDF) and a medium content of energy (2.5 Mcal ME/kg DM) [21,22]. Holstein Friesian heifers require about 12–16% CP and 2.5–2.6 Mcal ME/d [12] to reach the minimum live weight before mating. Therefore, Bv could be used to satisfy nutritional requirements and productive targets in growing Holstein heifers. Despite the high nutritive value of Bv, several studies have reported an unbalance in its energy and protein content [21,22], which could reduce the rumen N utilization and increase N excretion into the environment [21]. Although there are studies reporting the good growing potential and nutritive value of Bv in countries such as Chile [22,23] and New Zealand [20], there is a lack of studies evaluating the impact of Bv on animal performance and measures of sustainability such as N excretions. Thus, the aim of the study was to evaluate the effect of grazing *Lolium perenne* and *Bromus valdivianus* on average daily weight and nitrogen use efficiency of Holstein Friesian heifers.

## 2. Materials and Methods

### 2.1. Experimental Design and Treatments

The grazing experiment was conducted at the "Vista Alegre" Research Station of the Universidad Austral de Chile (39°47' S, 73°13' W, Valdivia, Chile) over a period of 30 days (October–November 2020), where the first 20 days corresponded to diet adaptation, with the last 10 days comprising the experimental sampling period. The climate in this area is temperate and humid, characterized by an average rainfall of 1754 mm/year, typically concentrated in the winter months, with an average air temperature of 11.2 °C. The trial was performed on Typic Hapludand soil, with an initial water pH of 5.7, Olsen-P of 11.2 mg/kg, exchangeable potassium of 142 mg/kg and 10% aluminum saturation. All the chemical features of the soil were measured for the first 20 cm of the soil profile. All animal experimental procedures were approved by the Animal Welfare Committee of Universidad Austral de Chile (grant number 392/2020).

A randomized complete block design with 2 treatments arranged in 3 field blocks was used in the study. Thirty Holstein Friesian heifers were grouped (n = 5) by body weight (BW; 344 ± 16 kg) and body condition score (2.78 ± 0.23, 0–5 scale). Heifers within groups were randomly allocated to one of two pasture treatments: (1) heifers grazing an Lp pasture and (2) heifers grazing a Bv pasture. Therefore, each treatment (n = 15) was composed by 3 groups of 5 heifers.

## 2.2. Pasture Preparation and Grazing Management

The experiment was conducted on a 3 ha paddock that was divided into six 0.5 ha plots arranged in three field blocks. Each block contained two 0.5 ha plots, one with Lp and the other with Bv. Prior to the establishment of the pastures (Lp and Bv), the areas were sprayed with glyphosate (N-(Phosphonomethyl)glycine) using a dosage of 2025 g/ha of the active ingredient, 60 days prior to sowing time. Limestone (tyler mesh +M100 < 10%) was applied to the surface 30 days prior to sowing at a dosage of 2000 kg $CaCO_3$/ha to correct aluminum saturation and soil pH, and then incorporated with tillage. The Lp and Bv pastures were sown in cultivated soil in March 2020 with 25 kg/ha certified cv. Stellar (AR1, >90% of seeds containing the endophyte) and 40 kg/ha cv. Bronco for Lp and Bv, respectively. Fertilizers were incorporated at sowing (25 kg N/ha, 120 kg $P_2O_5$/ha, 100 kg $K_2O$/ha). Two applications at a rate of 30 kg N/ha of urea were applied in August and September.

Each plot (200 m × 25 m) was strip-grazed by Holstein Friesian heifers once pastures were well-established. All plots of each treatment were simultaneously grazed for a period of 10 h. Animals began to graze when the pastures reached an average pre-grazing herbage mass of 2400–2600 kg DM/ha during spring and autumn or 1800–2000 kg DM/ha during winter. When a plot pasture did not reach the pre-grazing herbage mass criteria, the plots were grazed 60 days after the previous grazing event.

In mid-August, pastures were strip-grazed, and the subsequent regrowth was left for the current experiment. Each grazing strip was divided by electric fences. The daily area was adjusted by pre-grazing herbage mass (2400–2600 kg DM/ha; above ground level) and estimated daily DM intake (DMI; 2.5% BW). Pre- and post-grazing herbage mass was estimated daily from 100 compressed sward height measurements using a rising plate meter (Ashgrove Plate Meter, Hamilton, New Zealand). Compressed height data (cm) were transformed into kg DM/ha, using specific equations developed previously for each pasture species:

$$Lp: Y = 95*X + 21; r^2 = 0.831$$

$$Bv: Y = 92*X + 399; r^2 = 0.823$$

where: Y = herbage mass (kg DM/ha) and X = average compressed height (1/2 cm).

## 2.3. Forage Sampling and Analysis

Pasture samples were collected on day 7, 14, 21 and 28 of the experiment, before heifers had access to the new daily strip. Samples were composed by 5 pasture sub-samplings, which were collected from different places of the daily pasture allocation. Samples were collected manually by simulating the bite size, plant species and intake behavior of animals, using the hand-plucking technique [24]. Therefore, the pasture was cut approximately at 5 cm above ground level. Pasture samples were immediately frozen for subsequent analysis. All samples were freeze-dried and ground through a 1 mm screen (Willey Mill, 158 Arthur H, Thomas, Philadelphia, PA, USA), and analyzed for DM, CP, acid detergent fiber (ADF), ash [25], neutral detergent fiber (NDF) [26], soluble protein (SP) [27], water-soluble carbohydrates using the anthrone-sulfuric acid reagent method [28] and metabolizable energy (ME) [29,30]. Chemical composition was determined at the Animal Nutrition Laboratory, at the Universidad Austral de Chile.

## 2.4. Dry Matter Intake and Body Weight

Pasture DMI was estimated using the indigestible marker technique [31] between days 14 and 28 of the trial. Heifers were supplemented with a paper capsule containing chromium oxide (Cr; 15 g/d, 99.9% wt/wt) using an oral dispenser in the morning prior to getting into the new pasture allocation and in the evening before heifers were housed. To determine total DMI, feces samples were collected at 08:00 h, 11:00 h, 14:00 h and 17:00 h [32] between days 25 to 28 of the experiment (4 days), and immediately frozen to −20 °C. Fecal samples were thawed, freeze-dried and analyzed for Cr by atomic absorption spectroscopy (Spectronic Genesys 5 spectrophotometer, Milton Roy, Ivyland, PA, USA) and ADF.

Pasture DMI was estimated using the following equation:

$$\text{Pasture DMI (kg DM/animal/d)} = \frac{\text{FE (kg DM/d)}}{(1 - \text{DMD})}$$

where FE is fecal excretion per day (kg DM/d), DMD = digestibility of DM.

Fecal excretion was estimated using the equation:

$$\text{FE (g DM/cow/d)} = \frac{\text{Cr intake (g/d)}}{\text{Cr content in faeces (g Cr/g DM)}}$$

$$\text{DMD (g/kg of DM)} = 100 - (100*[\text{ADFfeed}]/[\text{ADFfeces}])$$

Body weight (BW) of the heifers was measured before morning feeding on days $-1$, 0, 1, 29, 30 and 31 of the experiment, using a Gallagher Animal Scale (TWR-1 G02602). Initial and final body weight was calculated as the mean value of the three days. Total weight gain was determined by subtracting the initial weight from the final weight. Average daily gain was determined by dividing the total weight gain by the number of experimental days (30).

### 2.5. Urine and Ruminal Samples

Urine samples were collected via voluntary excretion or massaging the vulva to stimulate the urination. Approximately 40 mL of urine was collected at 08:00 and 12:00 on day 28, as well as at 16:00 and 20:00 h on day 29 of the experiment, with the aim of capturing daily variation in the N excretion. All samples were acidified with sulfuric acid (10% *v/v*) to minimize volatilization and immediately frozen ($-20\ ^\circ$C) up to the time of analyses. Urine samples were thawed and analyzed for N content using a N autoanalyzer (LECO FP528) based on the DUMAS method. Additionally, urine samples were used to determine purine derivatives (PDs; allantoin and uric acid) and creatinine by HPLC. Urine volume was estimated using creatinine concentration as a marker and assuming a daily creatinine excretion of 26 mg/kg of BW [33]. Urine volume was used to estimate total PD excretion and urinary N. The microbial protein synthesis (g/day) was calculated from the PD excretion, using equations reported by Chen and Orskov [34]:

$$\text{PDa} = \frac{\left(\text{PDex} - \left(0.385 \times \text{BW}^{0.75}\right)\right)}{0.85}$$

$$\text{MN} = \frac{(\text{PDa}*70)}{(0.83 * 0.116 * 1000)}$$

where PDa = PD absorbed; Pdex = PD excreted; BW = body weight (kg); MN = microbial N, g N/d.

Rumen fluid samples were collected by stomach tubing (Flora Rumen Scoop; Prof-Products, Guelph, ON, Canada). A 10 mL ruminal sample was taken at 08:00 and 12:00 h on days 28 and 29 of experiment. All samples were mixed with 0.2 mL of 50% sulfuric acid, and stored at $-20\ ^\circ$C pending determination of $NH_3$ by colorimetry [35].

### 2.6. Nitrogen Balance

Nitrogen intake and its partitioning into urine and dung were calculated using the DMI and N content of forage and excreta. The following equation were used:

$$\text{N intake (g N/d)} = \text{DMI(kg DM/d)} * \frac{\%\,\text{N forage}}{100}$$

$$\text{Urinary N (g N/d )} = \text{Urine (L)} * \frac{\%\,\text{N in urine}}{100}$$

$$\text{Faecal N (g N/d)} = \text{Faecal DM excretion} * \frac{\%\,\text{N in feces}}{100}$$

$$\text{Retained N (g N/d)} = \text{N intake} - (\text{Urinary N} + \text{Faecal N})$$

$$\text{NUE (\%)} = \frac{\text{Retained N}}{\text{N intake}} * 100$$

### 2.7. Statistical Analysis

Effects of treatments on body weight, average daily gain, DMI and nitrogen partitioning were analyzed using the MIXED procedure of SAS [36]. The model included the fixed effects of pasture treatment and random effects of the field block and group of animals.

Chemical composition of pasture was analyzed using the PROC MIXED of SAS. The model included the fixed effects of treatment, random effect of field block, day of sampling as a repeated measurement and interaction between treatment and time of sampling.

Assumptions of normality and homogeneity residuals were checked graphically (plots of residuals versus fitted values and normal quantile plots). Comparison between treatments was carried out using the Tukey test. Results were considered significant at $p < 0.05$ and tendency at $p < 0.10$.

## 3. Results

### 3.1. Chemical Composition of Pasture and Herbage Mass

Chemical composition of *L. perenne* and *B. valdivianus* are presented in Table 1. Dry matter concentration was 2.3 percentage points greater for Lp compared with Bv ($p < 0.05$), while CP tended to be 1.3 percentage points greater for Bv ($p = 0.07$). Soluble protein was not modified by pasture treatments ($p > 0.05$), averaging 6.0%. Concentration of NDF and ADF were 5.1 and 3.5 percentage pointsgreater for Bv compared to Lp, respectively ($p < 0.05$). Metabolizable energy and WSC were 2% and 27% greater for Lp compared to Bv, respectively ($p < 0.05$). Differences in the CP and WSC content of the pasture were reflected in the WSC/CP ratio, which was 38% greater for Lp ($p < 0.05$).

**Table 1.** Chemical composition of *Lolium perenne* and *Bromus valdivianus* during the experiment.

| Parameters [1] | Treatment | | SEM [2] | *p* Value | | |
|---|---|---|---|---|---|---|
| | *L. perenne* | *B. valdivianus* | | **Species** | **Week** | **Interaction** |
| DM | 18.8 | 16.5 | 0.24 | <0.01 | <0.01 | 0.26 |
| CP | 15.5 | 16.8 | 0.62 | 0.07 | 0.19 | 0.07 |
| SP | 6.0 | 5.8 | 0.24 | 0.71 | 0.79 | 0.56 |
| NDF | 54.7 | 59.8 | 0.82 | <0.01 | 0.13 | 0.56 |
| ADF | 30.5 | 34.0 | 0.53 | <0.01 | 0.22 | 0.16 |
| ME | 2.74 | 2.68 | 0.01 | <0.01 | 0.03 | 0.76 |
| WSC | 13.3 | 10.5 | 0.37 | <0.01 | 0.18 | 0.93 |
| WSC:CP ratio | 0.88 | 0.64 | 0.05 | <0.01 | 0.15 | 0.14 |
| Pre grazing HM, kg DM/ha | 3073 | 3149 | 114.3 | 0.65 | <0.01 | 0.88 |
| Post-grazing HM, kg DM/ha | 1220 | 1406 | 124.2 | 0.17 | 0.62 | 0.07 |

[1] DM: dry matter, CP: crude protein; SP: soluble protein; NDF: neutral detergent fiber; ADF: acid detergent fiber; ME: metabolizable energy (Mcal ME/kg DM); WSC: water-soluble carbohydrates; HM: herbage mass; [2] standard error of mean.

Pre- grazing and post-grazing herbage mass were similar between treatments, averaging 3111 and 1313 kg DM/ha, respectively ($p > 0.05$).

### 3.2. Dry Matter Intake and Body Weight

Effects of pasture treatments on DMI, nutrient intake, and body weight are presented in Table 2. Dry matter intake was unaffected ($p > 0.10$), whereas NDF intake tended to be greater for Bv ($p = 0.09$) and ADF intake was 15% greater for heifers grazing Bv ($p < 0.05$). ME intake was similar among treatments, averaging 18.14 Mcal ME/d.

**Table 2.** Dry matter intake, body weight and average daily gain (ADG) of heifers grazing *Bromus valdivianus* and *Lolium perenne*.

| Parameters [1] | Treatment | | SEM [2] | *p*-Value |
| --- | --- | --- | --- | --- |
| | *L. perenne* | *B. valdivianus* | | |
| Intake, kg DM/d | | | | |
| DMI | 6.82 | 6.88 | 0.47 | 0.69 |
| SP | 0.39 | 0.45 | 0.07 | 0.58 |
| ME | 18.02 | 18.26 | 1.52 | 0.82 |
| NDF | 3.78 | 4.25 | 0.24 | 0.09 |
| ADF | 2.09 | 2.39 | 0.12 | 0.03 |
| WSC | 0.80 | 0.70 | 0.10 | 0.47 |
| Body Weight, kg | | | | |
| Initial | 343.9 | 345.4 | 1.42 | 0.57 |
| Final | 373.6 | 376.5 | 2.19 | 0.71 |
| ADG, kg/d | 0.98 | 1.00 | 0.09 | 0.79 |
| Feeding conversion, kg/kg | 7.22 | 6.98 | 0.97 | 0.84 |

[1] DMI: Dry matter intake; SP: Soluble protein; ME: Metabolizable energy (Mcal ME/d); NDF: Neutral detergent fiber; ADF: Acid detergent fiber; ADG: Average daily gain; [2] Standard Error of the Mean.

Initial and final BW, ADG (kg/d), and feeding conversion did not differ between treatments ($p > 0.05$), averaging 345 kg, 375 kg, 0.99 kg/d, and 7.1 kg/kg, respectively.

### 3.3. Purine Derivatives, Microbial N Synthesis and Nitrogen Partitioning

The effects of pasture treatments on purine derivatives and microbial N synthesis are presented in Table 3. Allantoin, uric acid, total PD and absorbed PD were not modified by pasture treatments ($p > 0.05$), averaging 162 mmol/d, 4 mmol/d, 166 mmol/d and 152 mmol/d, respectively. Similarly, microbial N was unaffected by treatments ($p > 0.05$), ranging between 106 and 115 g N/d. However, the ruminal $NH_3$ concentration was 49% greater in heifers grazing Bv compared with those grazing Lp ($p < 0.01$).

**Table 3.** Effect of *Lolium perenne* and *Bromus valdivianus* on purine derivates (allantoin, uric acid, total and absorbed), microbial nitrogen and ruminal ammonia of grazing dairy heifers.

| Parameters [1] | Treatment | | SEM [2] | *p*-Value |
| --- | --- | --- | --- | --- |
| | *L. perenne* | *B. valdivianus* | | |
| Allantoin, mmol/d | 156.45 | 167.46 | 4.97 | 0.23 |
| Uric Acid, mmol/d | 3.63 | 3.46 | 0.21 | 0.64 |
| Total PD, mmol/d | 160.08 | 170.92 | 5.09 | 0.25 |
| Absorbed PD, mmol/d | 145.82 | 157.59 | 6.18 | 0.31 |
| Microbial N, g N/d | 106.01 | 114.57 | 4.50 | 0.31 |
| Ruminal $NH_3$, mmol/L | 6.50 | 9.70 | 0.48 | <0.01 |
| Nitrogen intake, g N/d | 204.8 | 235.8 | 8.9 | 0.08 |
| Urine N excretion, g N/d | 75.1 | 94.8 | 4.1 | <0.01 |
| Dung N excretion, g N/d | 80.9 | 86.6 | 2.3 | 0.23 |
| Total N excretion, g N/d | 156.0 | 182.3 | 4.6 | <0.01 |
| Retained N, g N/d | 49.8 | 46.6 | 4.0 | 0.25 |
| NUE, % | 23.2 | 20.7 | 2.8 | 0.93 |

[1] PD: purine derivatives; N: nitrogen; $NH_3$: ammonia; N: nitrogen; NUE: nitrogen use efficiency; [2] standard error of the mean.

Heifers that grazed Bv showed a tendency towards greater (+15%) N intake when compared to Lp ($p = 0.08$). Urinary and total N excretion were 26% and 17% greater for Bv compared to Lp ($p < 0.05$). However, dung N excretion did not differ between treatments, averaging 84 g N/d ($p > 0.05$). Retained N was not affected by treatments ($p > 0.05$). Despite differences in the N excretion, NUE was similar between treatments, averaging 22% ($p > 0.05$).

## 4. Discussion

The majority of adaptation and mitigation efforts to reduce the effects of climate change on dairy cattle have been related to grazing dairy cows, due to its importance for the economic stability and sustainability of the dairy farm [16]. There is scarce information on nutritional strategies to cope with climate change (adaptation) and in turn, reduce the N partitioning and excretion of pasture-based dairy heifers (mitigation of climate change). Furthermore, the current study evaluated growth performance and N partitioning of heifers grazing a traditional pasture species (Lp) and a promising native species, Bv, which tolerates the summer soil water restriction better than Lp, being an alternative as climate change increases the likelihood of drought during summer and consequently lowers its growth rates [37].

### 4.1. Chemical Composition of Pasture and Herbage Mass

Quality and quantity of pasture production must be considered when a new or novel pasture species is evaluated for inclusion in the grazing system of dairy cows, due to their impact on productive parameters such as growth or milk production [38,39]. In the current experiment, several factors related to pasture availability (pre- and post-grazing herbage mass) and quality in terms of nitrogen (CP) were similar between Bv and Lp, suggesting that Bv could be used to replace Lp, especially for the months of the year where the water deficit is increased. *Bromus valdivianus* is a native grass species from Southern Chile that is characterized by its high DM yield and good forage quality, even under restricted conditions of soil water content [40–42], which is related to its deep root system, which enables capture of water from deeper soil strata, increasing its drought tolerance [18–20]. It was observed that ADF and NDF content were greater for Bv, while WSC content was greater for Lp. Similar results have been previously reported in the literature [22,23], where WSC and NDF contents were greater for Lp and Bv, respectively.

### 4.2. Dry Matter Intake and Body Weight

Several factors, such as the environment, plant, animal, and management, determine the pasture intake in grazing-based systems [43]. In the current experiment, pasture DMI was similar between treatments, suggesting that Bv offers a similar chemical and physical composition to Lp, a traditional species used for pasture-based dairy systems. *Bromus valdivianus* presented a similar DMI and chemical composition to Lp during the time of the year where temperature and precipitation were adequate for pasture growing, suggesting that during the dry season (summer), where Lp shows a reduced pasture production, it could produce a high-quality pasture, as reported Alfaro et al. [21]. One of the most important consequences of climate change in cattle production is the reduction in pasture and crop availability for animal feeding [1,5]. Therefore, pasture species adapted to these conditions (such as pasture brome) are required. Additionally, pasture brome has shown to have similar $N_2O$ and carbon (expressed as $CO_2$-eq) emissions to pasture dominated by perennial ryegrass [21], supporting its importance in the adaptation to climate change. Therefore, in terms of similar pasture DMI and chemical composition, Bv could be considered as promising species adapted to climate change.

Even though N intake tended to be greater for heifers grazing Bv, body weight was similar among treatments, which could be related to the similar intake of DM and ME. According to Brown et al. [44], an increase in the energy and protein intake can increase the rate of body growth of heifer calves. However, the optimum effect was obtained in heifers receiving a high-protein diet. The effect of energy intake on ADG is relevant in protein-limited diets, which do not occur in grazing systems. Therefore, our results suggests that an increase in the N intake of 15% was not enough to evidence an effect in the ADG and thereby, the final BW. Conversely, an excess of N in the rumen increases the amount of $NH_3$ that is converted into urea in the liver. This detoxification process has an energy cost of 4 moles of ATP per mol of urea synthesized or 7.17 kcal ME per g N synthesized as urea [13], and thus it may reduce energy available for growth.

### 4.3. Nitrogen Metabolism and Partitioning

Pasture brome tended to have a greater CP concentration than perennial ryegrass; consequently, N intake tended to be 15% greater for heifers grazing Bv, which explains in part the greater ruminal $NH_3$ concentration. According to Ueda et al. [45], when N intake is increased, $NH_3$-N utilization by ruminal microbes can be improved when the supply of readily fermentable carbohydrates in the rumen is increased. Heifers grazing Bv had a high N intake, while their energy intake (ME and WSC) was similar to that of heifers grazing Lp; therefore, the excess of N was converted to $NH_3$ by ruminal bacteria [46], as a result of the imbalance in the supply of energy and protein in the rumen, which is supported by the greater N intake and lower WSC:CP ratio of Bv.

Despite differences in N intake between treatments, microbial N (g N/d) was not affected by treatment, reflecting that the increase in N intake was not utilized for synthesis of microbial N, and instead an increased rumen N concentration [45]. In fact, energy supply is the main factor limiting microbial growth in the rumen of cattle grazing temperate pastures [47,48]. According to Arias et al. [49], a lower $NH_3$ concentration in the rumen is associated with higher $NH_3$ utilization by ruminal microbial protein. However, this was not observed herein. Instead, we observed a greater ruminal $NH_3$ concentration in heifers grazing Bv, but no relation with lower microbial N in the rumen. This indicates that both treatments supplied the required energy and protein to produce the same microbial N; therefore, the excess N intake for heifers grazing Bv was transported into the liver to be converted into urea and then excreted through urine or recycled with saliva [14].

Retained N was not modified by treatments, suggesting that both types of pasture supplied the required N for heifers. Although Lp showed a greater WSC:CP ratio, there was no difference in the NUE of heifers grazing pasture brome and those grazing perennial ryegrass. It has been reported that ruminal utilization of N by microorganisms increases as the WSC:CP ratio increases in the diet, in response to a greater supply of readily fermentable carbohydrates in the rumen [50,51]. This is an important result, which reflects that NUE is a limited tool in estimating potential N pollution, as it only determines the fraction of ingested N that is not retained by the animals and does not consider the amount of N excreted in urine and feces. For this reason, NUE must be analyzed along with the N partitioning into urine and dung to determine the real effect of treatments on N utilization by the animal. For example, Beltran et al. [11] observed that NUE was greater for dairy than beef cattle; however, the urinary N excretion (g N/d) was lower for beef than for dairy cattle.

The trend towards a greater N intake may explain the greater urinary N excretion in Bv fed heifers, as N intake has been described as the main factor determining urinary N excretion [50,52], with a linear relationship between them. The surplus of N intake compared with requirements is usually excreted into the environment [13]. The greater urinary N excretion is in part the consequence of greater concentrations of $NH_3$ in the rumen. Once $NH_3$ is produced by ruminal bacteria, it can be used to build microbial protein (an energy-dependent process) or can be transported to the liver (low energy availability in the rumen) to be converted into urea and then may be excreted through the urine [13,14,46] or recycled into the rumen along with saliva [53]. A positive relationship between ruminal $NH_3$ and urinary N excretion has been described for grazing dairy cows [46], similar to our results. Although Alfaro et al. [21] suggested that Bv has the potential to combat climate change (adaptation to high temperatures and low precipitations), the imbalance in its N and energy content may increase the urinary N excretion into the environment. Nitrogen excretion through the urine and dung is important in terms of environmental pollution, because they are an important source of N for $N_2O$ emissions in pasture-based livestock systems, a powerful greenhouse gas whose global warming potential is 265 times greater than that of carbon dioxide [7]. Nitrous oxide emissions from urine are 5 times greater than those of dung N; therefore, the focus of nutritional strategies should shift to the N excretion from urine to dung or to reducing urinary N excretion, with the aim of reducing $N_2O$ [52]. Therefore, the greater urinary N excretion by heifers grazing Bv may result in greater $N_2O$ emissions compared to heifers grazing Lp.

Dung N excretion was not affected by treatments, which was an expected result as dung N is mainly composed of indigestible N in the diet, which does not show a linear relationship with N intake, suggesting that other parameters of the diet must be modified to produce an increase in the dung N excretion, such as rumen degradation parameters. However, this may not be the case as Keim et al. [38] observed similar in situ effective rumen degradability values for pastures containing 90% perennial ryegrass or 48% pasture brome and 43% perennial ryegrass, suggesting that inclusion of Bv in the pasture may not affect in situ rumen degradation parameters.

### 4.4. Implications of the Study

The impact of climate change on agriculture is expected to restrict animal access to pasture and crop production for animal feeding. Therefore, it is important to identify pasture species adapted to high temperatures and low precipitations. Our study showed that Bv and Lp had similar pasture characteristics in terms of availability (pre-grazing herbage mass) and quality (CP, ME and NDF) during the spring season, suggesting that Bv could be used during the time of year when Lp reduces the pasture production and composition, i.e., during the summer, which would allow for extending the grazing season. *Bromus valdivianus* could be used as an adaptation strategy in the face of climate change through its combination with Lp, thus reducing GHG emissions ($N_2O$). Previously, it has been shown that mixtures dominated by perennial ryegrass and pasture brome can have similar attributes to Lp pastures in terms of DM yield [54] and fermentation in the rumen [38,39]. More recently, García-Favre et al. [37] observed a synergy when combining Lp and Bv by increasing DM yield by 15% compared with Lp, mainly due to an increase in production in spring and summer. In spring, there was a complementarity growth between both species, while during summer/early autumn, the production resulted in the higher participation of Bv as well as a greater root biomass at a depth of 31–70 cm.

The main concern related to the use of Bv shown in this study was the increase in the urinary N excretion, because of its greater N intake (without increasing the WSC intake), which reduced the ruminal N utilization and thereby increased the N excretion in the urine. Urinary N is the main source of $N_2O$ emissions from grazing systems; therefore, we could expect greater urine and dung $N_2O$ emissions from heifers grazing Bv compared to heifers grazing traditional pasture dominated by Lp. In this way, it is necessary to evaluate if adaptation to climate change offsets its suggested increase in $N_2O$ emissions, which could exacerbate global warming. One limitation of this study is that it was conducted for a limited period of time, during one growing season (spring). In humid temperate regions such as Southern Chile, more than 50% of total annual herbage mass of grass pastures is produced during spring [22,39]; thus, spring was selected as the time period for this experiment. Although chemical composition varies between spring and autumn (with greater CP and lower WSC in autumn), no differences compared with the results from this study should be expected between Lp and Bv in autumn, as seasonal changes in the chemical composition of the two species occur in a similar pattern [23]. Additionally, research on the persistence and the use of Bv throughout the year is required to understand its impact on the forage system and the nitrogen balance (uptake/emission) in a heifer grazing system.

### 5. Conclusions

The inclusion of Bv in the diet of Holstein Friesian heifers allowed us to maintain the NUE and ADG when compared to Lp, mainly due to the similar pasture chemical composition and intake. This suggests that Bv can be included in the grazing season, especially during the time of the year when high temperatures and low precipitations reduce pasture production from Lp, offering a solution to cope with climate change effects on pasture-based systems. However, it must be considered that Bv increased the urinary N excretion of heifers, which may increase the environmental pollution from cattle. Therefore, the persistence of Bv along with a full grazing season evaluation should be conducted to

identify its impact on pasture yield, animal production and nitrogen balance in a heifer grazing system.

**Author Contributions:** Conceptualization, O.B. and J.P.K.; methodology, J.P.K., O.B. and R.A.; software, J.P.K. and D.T.; validation, O.B., R.A., J.P.K. and I.E.B.; formal analysis, O.B., R.A., J.P.K. and I.E.B.; investigation, D.T., J.C., J.P.K.; resources, O.B. and J.P.K.; data curation, J.P.K., D.T. and J.C.; writing—original draft preparation, D.T. and I.E.B.; writing—review and editing, J.P.K., O.B. and R.A.; visualization, I.E.B. and J.P.K.; supervision, J.P.K.; project administration, O.B. and J.P.K.; funding acquisition, O.B. All authors have read and agreed to the published version of the manuscript.

**Funding:** This research was funded by FONDECYT, grant number "1180767".

**Institutional Review Board Statement:** All animal experimental procedures were approved by the Animal Welfare Committee of Universidad Austral de Chile (grant number 392/2020).

**Data Availability Statement:** The data presented in this study are available on request from the corresponding author.

**Acknowledgments:** Authors would like to acknowledge the staff of the Austral Research Station for their collaboration with crop production, animal handling and feeding.

**Conflicts of Interest:** The authors declare no conflict of interest.

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
