# Peer review of "Nitrogen Use Efficiency and Partitioning of Dairy Heifers Grazing Perennial Ryegrass (Lolium perenne L.) or Pasture Brome (Bromus valdivianus Phil.) Swards during Spring"

_agronomy, doi:10.3390/agronomy12081953_

Round 1

Reviewer 1 Report

I appreciate the hard work in collecting data.

Author Response

Thanks for your recognition

Reviewer 2 Report

Line 167 : need to be corrected "on day and 29"

Table 1 : need to be corrected bold front and underlined for "DM"

Table 2 : need to be corrected bold front and underlined for "Intake, kgDM/d
     and be aligned "0.47" (SEM value for DMI)

Table 3 : need to be corrected bold front and underlined for "L.perenne" and "Allantoin, nmol/d" and "156.45"

Line 336 : need to be corrected "towards a greater N intake"

Line 397 and conclusion : recommendation to expand the conclusion by adding the need for further research on the use of Bv over a full grazing season, its place in a forage system and the nitrogen balance (uptake/emission) in a heifer grazing system.

Author Response

Line 167 : need to be corrected "on day and 29"

  • Answer: Text was corrected as suggested by reviewer.

Table 1 : need to be corrected bold front and underlined for "DM"

  • Answer: Table was corrected as suggested by reviewer.

Table 2 : need to be corrected bold front and underlined for "Intake, kgDM/d
     and be aligned "0.47" (SEM value for DMI)

  • Answer: Table was corrected as suggested by reviewer.

Table 3 : need to be corrected bold front and underlined for "L.perenne" and "Allantoin, nmol/d" and "156.45"

  • Answer: Table was corrected as suggested by reviewer

Line 336 : need to be corrected "towards a greater N intake"

  • Answer: Text was corrected as suggested by reviewer.

Line 397 and conclusion: recommendation to expand the conclusion by adding the need for further research on the use of Bv over a full grazing season, its place in a forage system and the nitrogen balance (uptake/emission) in a heifer grazing system.

  • Answer: A sentence was included at the end of “Implications of the study” and “Conclusion” sections. Check L400-402 and L411-412.

Reviewer 3 Report

The paper reports an interesting study on nitrogen use efficiency and excretion of grazing dairy heifers grazing two different grass species.  I recommend its publication.  Some moderate changes are needed to English usage. The following are suggested improvements by line (L) number.  Some greater clarity also is needed particularly with explaining some of the methodologies used in the experiment.

L22.  Delete “were not affected by grass species”.  L21 states the same.

L26.  Change to “compared with Lp”.

L35.  Change to “consequences”.

L71.  Delete “of”.

L105.  Add the chemical formulation in parentheses after glyphosate and state how much time before establishment of the pastures glyphosate was applied.

L106.  State how limestone was applied and when.  Was it incorporated into soil with any tillage?  Also, how fine or coarse of limestone was used?

L129.  Change “Food” to “Forage”.

L133.  What is the ground area that was sampled?  The hand-plucking technique is unclear. 

L142.  What was done to get the heifers to consume the marker?  Were the heifers removed from the pasture for dispensing the marker?  Greater explanation of the technique is needed.

L144.  The methodology of how fecal samples were collected needs greater explanation. 

L165.  Please provide more explanation to how urine was captured or collected from the animals.  It is unclear the meaning of “urine spot”.

L167. Delete “and” between “day” and “29”.

L188.  Replace “food” with “forage” and use consistent terminology in the equation in L190, assuming diet is synonymous with forage.

L209. Replace “content” with “concentration”. Then state actual percentage points rather than % differences, e.g., 2.3 and 1.3 percentage points for DM and CP, respectively, from here on out.  The percentage differences do not compute.  For DM, it would be 14% greater for Lp than Bv.   For CP, it would be 8.3% greater for Bv than Lp.  

L210.  How was soluble protein determined?  It was not explained in the Materials & Methods.

L212.  How was WSC determined?  It also was not explained.

L272.  Consider replacing “allows to obtain water” with “enables capture of water”.

L273.  Are the “similar” results if you observed NDF greater for Bv, but the cited references found greater NDF in Lp?

L291.  The sentence ends abruptly and lacks clarity.  Please revise.

L322.  Revise to “the excess N intake of heifers grazing Bv”.

L336. Fix spelling of the first “greater”.

L351. End sentence with the reference after “systems”.  The sentence, otherwise, ends abruptly.

L366. The sentence is unclear.  I’m not sure “optimize” is the best verb here.  Consider revising this statement.

L371. Revise to “time of year when Lp has reduced pasture production, i.e., during summer, which would allow an increase in the grazing season”.

L377. Revise to “when combining Lp and Bv, increasing DM yield by 15%”.

L394. Replace “were” with “when”.

Author Response

The paper reports an interesting study on nitrogen use efficiency and excretion of grazing dairy heifers grazing two different grass species.  I recommend its publication.  Some moderate changes are needed to English usage. The following are suggested improvements by line (L) number.  Some greater clarity also is needed particularly with explaining some of the methodologies used in the experiment.

Many thanks for your comments and suggestions.

L22.  Delete “were not affected by grass species”.  L21 states the same.

  • Answer: Text was corrected as suggested by reviewer

L26.  Change to “compared with Lp”.

  • Answer: Text was corrected as suggested by reviewer

L35.  Change to “consequences”.

  • Answer: Text was corrected as suggested by reviewer

L71.  Delete “of”.

  • Answer: Text was corrected as suggested by reviewer

L105.  Add the chemical formulation in parentheses after glyphosate

  • Answer: Chemical formulation of Glyphosate (N-(Phosphonomethyl)glycine) was included in the text. L105

and state how much time before establishment of the pastures glyphosate was applied.

  • Glyphosate was applied 60 days previous to sowing time. Please, check L106.

L106.  State how limestone was applied and when.  Was it incorporated into soil with any tillage?  Also, how fine or coarse of limestone was used?

A: Information included (please see lines 105-108)

L129.  Change “Food” to “Forage”.

  • Answer: Text was corrected as suggested by reviewer

L133.  What is the ground area that was sampled?  The hand-plucking technique is unclear. 

  • Answer: More details on hand-plucking technique was included in the text. Check L134-136.

L142.  What was done to get the heifers to consume the marker?  Were the heifers removed from the pasture for dispensing the marker?  Greater explanation of the technique is needed.

  • Answer: An explanation of indigestible marker technique was included in the text. Please, check L143-147.

L144.  The methodology of how fecal samples were collected needs greater explanation. 

  • Answer: An explanation of indigestible marker technique was included in the text. Please, check L147-150

L165.  Please provide more explanation to how urine was captured or collected from the animals.  It is unclear the meaning of “urine spot”.

  • Answer: We have deleted the word “spot” to avoid confusion in the readers. Additionally, we have included more details on urine sampling.

L167. Delete “and” between “day” and “29”.

  • Answer: Text was corrected as suggested by reviewer.

L188.  Replace “food” with “forage” and use consistent terminology in the equation in L190, assuming diet is synonymous with forage.

  • Answer: Text and equations were corrected as suggested by reviewer.

L209. Replace “content” with “concentration”. Then state actual percentage points rather than % differences, e.g., 2.3 and 1.3 percentage points for DM and CP, respectively, from here on out.  The percentage differences do not compute.  For DM, it would be 14% greater for Lp than Bv.   For CP, it would be 8.3% greater for Bv than Lp.  

  • Answer: Text and percentages expression were corrected as suggested by reviewer. Check L212-220.

L210.  How was soluble protein determined?  It was not explained in the Materials & Methods.

  • Answer: Included, according to licitra et al. (1996)

L212.  How was WSC determined?  It also was not explained.

  • Answer: Included, by the anthrone-sulphuric acid reagent method (MAFF, 1986)

L272.  Consider replacing “allows to obtain water” with “enables capture of water”.

Answer: Text was corrected as suggested by reviewer.

L273.  Are the “similar” results if you observed NDF greater for Bv, but the cited references found greater NDF in Lp?

  • We have checked the cited references, and our discussion is right. Calvache et al. (2020a; DOI: https://doi.org/10.3390/agronomy10050620 ) found that NDF was greater for Bv, while WSC was greater for Lp. Calvache et al. (2020b; DOI: https://doi.org/10.3390/agriculture10110563 ) found thar WSC was greater for Lp compared to Bv.

L291.  The sentence ends abruptly and lacks clarity.  Please revise.

  • Answer: Sentence was completed (298-299).

L322.  Revise to “the excess N intake of heifers grazing Bv”.

  • Text was edited as suggested by reviewer.

L336. Fix spelling of the first “greater”.

  • Answer: Text and percentages expression were corrected as suggested by reviewer.

L351. End sentence with the reference after “systems”.  The sentence, otherwise, ends abruptly.

  • Sentence was edited as suggested by reviewer.
  •  

L366. The sentence is unclear.  I’m not sure “optimize” is the best verb here.  Consider revising this statement.

  • Sentence was edited as suggested by reviewer.

L371. Revise to “time of year when Lp has reduced pasture production, i.e., during summer, which would allow an increase in the grazing season”.

  • Answer: Text was modified as suggested by reviewer

L377. Revise to “when combining Lp and Bv, increasing DM yield by 15%”.

  • Answer: Text was revised and corrected. Check L387.

L394. Replace “were” with “when”.

  • Text was corrected.